# The Circadian Modulators as Molecular Targets in Cancer—A Review

**DOI:** 10.3390/ijms262411779

**Published:** 2025-12-05

**Authors:** Anna Wolniakowska, Joanna Roszak, Zuzanna Sobańska, Edyta Reszka

**Affiliations:** 1Center for Alternative Methods for Toxicity Assessment, Nofer Institute of Occupational Medicine, Św. Teresy 8, 91-348 Łódź, Poland; anna.wolniakowska@imp.lodz.pl (A.W.); joanna.roszak@imp.lodz.pl (J.R.); zuzanna.sobanska@imp.lodz.pl (Z.S.); 2Department of Biophysics of Environmental Pollution, Faculty of Biology and Environmental Protection, University of Lodz, Pomorska 141/143, 90-236 Łódź, Poland

**Keywords:** small-molecule modulators, circadian clock, cancer

## Abstract

Disruptions in the circadian clock and their link with cancer constitute a growing area of research, as evidenced by the steadily increasing number of articles on the topic. While the genes associated with circadian rhythms are relatively well characterized, the complexity of their regulation remains an important direction for study. It has been demonstrated that the interplay between genes, their expression products and external factors, such as environmental pollutants and human behavioral patterns, can lead to pathological changes, including metabolic diseases and cancer. Investigation of circadian cycle deregulations can not only provide a better understanding of carcinogenicity mechanisms and risk assessment but also create possibilities to identify new chemotherapeutics targeted at neoplastic cells. REV-ERBs and RORs are two groups of circadian clock-related nuclear factors that are examined regarding their interactions with small-molecule modulators of the circadian clock. These can act as either receptor agonists or inverse agonists, depending on the specific characteristics of a particular cancer. This review therefore summarizes and systematizes existing knowledge regarding the effectiveness of circadian modulators as chemotherapy agents, with the aim of indicating further directions for research in the field.

## 1. Introduction

The correct functioning of living organisms is in many ways determined by their circadian clock. This system plays a fundamental role in regulating daily changes in physiology and behavior, as well as responses to various environmental and metabolic stimuli, through a complex internal system of molecular pathways [1]. The growing body of knowledge regarding the impact of the circadian clock on the human body has resulted in the development of chronobiology. While the terms circadian disruption or circadian misalignment have been used since the 1980s, they only gained publicity in 2001 with the recognition of the link between nighttime lighting, endocrine disorders and breast cancer risk [2]; both terms indicate disruptions caused by either genetic, environmental or internal factors [3]. Regardless of the cause, circadian disturbances are also believed to play roles in various conditions, including cardiovascular and metabolic diseases (diabetes, obesity), depression and cancer [4].

The mammalian circadian clock, also known as the circadian rhythm, is composed of central and peripheral clocks creating interconnected transcription and translational feedback loops (TTFLs) that involve several genes and their proteins with an intrinsic period that is close to 24 h [5]. Briefly, the central clock is the master pacemaker of circadian rhythms, and it is located in the hypothalamic suprachiasmatic nucleus (SCN), which contains 10,000–15,000 neurons [6], while the peripheral clock pathways are found in various organs such as the liver, brain, lungs and heart.

The molecular clock constitutes a very complex network of motifs necessary for generating an independent and self-sustained circadian rhythm. The core molecular clock’s main players include, but are not limited to *CLOCK*, *ARNTL* (*BMAL1*), *PERIOD* (*PER*), *CRYPTOCHROME* (*CRY*), *REV-ERBs* (*NR1Ds*) and *RORs* (Table 1). They create interlocking feedback loops consequent to interaction between the positive regulators BMAL1 and CLOCK and the negative regulators CRY and PER (Figure 1). The BMAL1/CLOCK heterodimer induces transcription of target genes by binding to the E-box sequence in the promoter region, and it upregulates the transcription of PER and CRY. The PER and CRY proteins form a heterodimer in the cytoplasm that is translocated into the nucleus, and it subsequently inhibits the activity of the BMAL1/CLOCK heterodimer. REV-ERB and ROR, two retinoic acid-related orphan nuclear receptors, constitute the stabilizing loop by competing to bind to the retinoic acid-related orphan receptor-binding element (RORE) in the *BMAL1* gene. During the transcription process of *BMAL1*, ROR activates it, but REV-ERB suppresses it [7]. Circadian oscillation of PERs with positive (DBP, TEF, and HLF) and negative (NFIL3) cis-elements is provided by the third D-box-driven auxiliary loop [8] (in Figure 1).

Post-translational regulation of circadian proteins is also essential for molecular clock maintenance. In the cytoplasm, PERs and CRYs undergo phosphorylation by casein kinases CSNK1E and CSNK1D. Accumulated clock proteins are then transported to the nucleus, where they form a protein complex that inhibits the expression of BMAL1 [8].

The peripheral circadian clock is responsible for controlling a vast and diverse level of transcripts, as indicated by several rodent and human data. The main mechanism of circadian control of gene expression relies mainly on the BMAL/CLOCK heterodimer, triggering the expression of a variety of clock-controlled genes (CCGs) with diverse effects on cell function and physiology, with possible implications for cancer in particular and health in general [9]. Therefore, clinical treatments for the targeting of clock genes may represent a promising approach for modulating tumor progression [10] or the immune system [11,12], or the antioxidant defense [13]. As all these mechanisms play important parts in novel drug development, this review will focus on the impact of circadian clock modulators on potential cancer chemotherapeutics.

## 2. Circadian Clock and Cancer

The molecular basis of cancer has been elucidated through intensive investigation of changes in clock gene alterations in cancer development. The core clock genes may demonstrate different levels of expression between different tissues [14]. For example, studies indicate that while *CRY1* is downregulated in osteosarcoma, it is upregulated in colorectal cancer [15]. Furthermore, *PER2* expression is decreased in breast, lung [16] and ovarian tumor tissues [17], but increased in gastric cancer tissues [18]. *REV-ERBβ* is upregulated in various tumor cell lines derived from breast, prostate, liver and skin [19].

The observation of dysregulation of circadian rhythm by both genetic and environmental factors pointed to a close connection between molecular clocks and cancer pathways. In 2007, the International Agency for Research on Cancer (IARC) recognized the circadian disruption related to night shift work as a risk factor for cancer [20]. However, mitigating this risk can present a challenge, as particular elements of the circadian clock may influence cancer initiation or progression through several mechanisms (Figure 2).

The circadian clock regulates gene expression in various tissues of the body, both directly and indirectly; as a result, it can influence a range of cellular processes, such as autophagy [21], cell cycle [22], DNA damage repair or protein folding [23]. Any disruption of the circadian rhythm also disturbs metabolic processes, which can lead to cell reprogramming, redox imbalance and chronic inflammation [24]; these contribute to forming a cellular environment that can promote the development of cancer. Moreover, many circadian clock proteins interact with molecules involved in tumorigenesis, and with the secretion of paracrine and endocrine factors, which support tumor metastasis [25].

Circadian variation is a common feature of xenobiotic metabolism [26]. Thus, the effectiveness and toxic side effects of different drugs can be affected by the time of day, which is a crucial aspect of chronotherapy in various diseases, including cancer [27]. Drug administration can have time-dependent effects on either pharmacokinetics (absorption, metabolism, or elimination) or pharmacodynamics (the response of the cell) [28].

The molecular clock can be modified by specific drugs, but identifying the precise small-molecule modulator of clock components in cancer therapy may be challenging. Additionally, the timing of administration may affect its pharmacological properties of small-molecule ligands (agonist, partial agonist, inverse agonist, antagonist) [29].

## 3. Circadian Modulators as Molecular Targets

The growing understanding of how circadian clock disruptions affect disease development has allowed for the development of new approaches in chronotherapy. Sulli et al. describe three possible pathways by which cancer development can be halted: Firstly, the clock can be trained by maintaining a solid circadian rhythm in the feeding-fasting, sleep–wake or light–dark cycle. Secondly, drug application can be “clocked” by optimizing the timing of drug administration to improve efficacy and reduce unwanted side effects. Thirdly, the clock can be “drugged” by using small molecules to directly target a component of the circadian clock or an associated factor [30].

Cancer treatment based on pharmacological modulation of clock components (drugging the clock) usually employs small-molecule compounds that act as modulators of their proteins; these constitute a rapidly growing area of research. The small molecules have been found to act on various cellular pathways that modulate molecular feedback loops, leading to changes in the amplitude, phase and period of circadian rhythms. In particular, research on deregulating the molecular clock focuses on natural small-molecule ligands that enhance clock element function, such as RORs and REV-ERBs receptors.

### 3.1. Modulators of REV-ERBs

REV-ERBs belong to a large superfamily of ligand-regulated transcription factors. Among these, REV-ERBα and REV-ERB β (encoded by *NR1D1* and *NR1D2*, respectively), are the main modulators of *BMAL1* expression and the auxiliary feedback loop in the clock cycle [31]. Changes in the proper functioning of these receptors may result in deregulation of the circadian rhythm and hence, the disruption of various physiological processes, such as the sleep–wake cycle, feeding and fasting rhythms, blood glucose and lipid levels.

While REV-ERBs were first proposed as targets for the treatment of sleep and metabolic disorders [32], it has since been found that these heme-binding factors in fact repress the cell metabolism, proliferation and inflammation processes involved in tumorigenesis [33]; they have also been found to be engaged in gastric cancer [34], breast cancer [19,35], colorectal cancer [36], lung adenocarcinoma [37] and thyroid cancer [38]. Most data indicates that REV-ERBs are downregulated in tumor tissue, which is associated with higher cell proliferation; as such, pharmacologically activating REV-ERBs might have positive therapeutic implications. The observation that REV-ERBs are ligand-dependent receptors began a search for synthetic small-molecule modulators witch different chemical structure (Figure 3) that could modulate their activity. Table 2 provides a summary of small-molecule REV-ERB ligands investigated in different anticancer therapies.

The first described small-molecule ligand for REV-ERBs was the REV-ERBα agonist GSK4112, also known as SR6452. The ligand competes with heme [48]. An in vitro study of human gastric adenocarcinoma cells (cell lines SGC-7901 and BGC-823) found GSK4112 treatment to reduce cell viability; cell death was attributed to ferroptosis caused by upregulation of REV-ERBα [34,39].

Building on the promising results of GSK4112 research, other small-molecule modulators were developed based on derivatives with better pharmacokinetics in rodents. Among these, the most frequently described GSK4112 analogs, i.e., those with a similar tertiary amine structural motif, are SR9009 and SR9011. Both were synthesized by reductive aminations via the activity of ethyl chloroformate (SR9009) and pentyl isocyanateon (SR9011) in the last step. Both compounds turned out to be selective, exhibiting activity only at REV-ERB α/β receptors, and were found to suppress the expression of *BMAL1* messenger RNA in a REV-ERB-α/β-dependent manner [32]. Both compounds were also observed to influence the metabolism and have been evaluated in metabolic diseases, such as obesity, as well as in sleep cycle disorders and psychiatric illnesses, e.g., anxiety and depressive disorders [32].

Both SR9009 and SR9011 effectively influence the circadian clock. Studies have found that they alter the physiological time of day of male C57BL6 mice, suggesting that they may be useful in treating sleep disorders. [32,49]. Treatment was also reported to influence weight loss in mice, mainly loss of fat mass; this loss probably resulted from increased basal oxygen consumption, rather than an increase in activity or a decrease in food intake [50]. SR9009 and SR9011 were also shown to modulate emotional behavior [49] and the immune response [51,52], which may be important in the development of novel drugs for treating anxiety, as well as metabolic and neuroinflammatory diseases.

Studies have also found SR9009 to exhibit antitumor effects against both chemosensitive (H69 and H446) and chemoresistant (H69AR and H446DDP) human small-cell lung cancer cells (SCLC) in vitro, as well as SCLC subcutaneous tumors in an in vivo mouse model. The antitumor effect of SR9009 in this case was attributed to autophagy inhibition, mediated by direct downregulation of the core autophagy gene *Atg5* by REV-ERB-α [40].

The ability of SR9009 to inhibit autophagy was also investigated by Wang et al. [41] as part of two experiments: one in vitro procedure based on human multiple myeloma (MM) cell lines (RPMI8226 and U266) and an in vivo procedure using a non-obese diabetic/severe combined immunodeficient (NOD/SCID) murine xenograft MM model. The in vitro model indicated reduced cell viability and proliferation rate due to a dose-dependent apoptotic response, while decreased tumor growth was noted in vivo [41].

SR9009 also demonstrated similar antiproliferative effects in bladder cancer (BC); downregulated *NR1D1* was associated with poor prognosis, and REV-ERB activation suppressed the invasion of BC cells in in vitro models (cell lines: RT4, T-24 and 5637) [42].

A study on hepatocellular carcinoma found SR9009 to not influence healthy hepatocytes; however, it impaired tumor cell growth and inhibited migration in a REV-ERB-dependent manner [43]. A study of hepatocellular carcinoma and glioma cells indicated that SR9009 treatment significantly altered tumor cell metabolism and promoted cytotoxic effects [44]. In addition, SR9009 was found to selectively decrease the viability of prostate cancer (PC) cells in a dose-dependent manner, while having no effect on normal prostate cells [45].

Research suggests that SR9011 may be of value in the treatment of diseases associated with uncontrolled cell proliferation. In vitro studies on breast cancer cell lines showed that suppression of cyclin A expression by SR9011 may lead to dose-dependent cell cycle arrest, thus preventing tumor growth [46]. SR9011 was also found to effectively inhibit the growth of malignant osteosarcoma cells in vitro and in vivo through the mTOR signaling pathway, but did not hinder the progression of normal osteoblasts [47].

Other compounds developed on the basis of GSK4112 scaffolds include GSK2945, GSK0999, GSK5072 and GSK2667 [53]. All show much higher specificity to REV-ERBs and greater bioavailability than the parent GSK4112. Among these, it is still unclear whether GSK2945 acts as an agonist or an antagonist; this ambiguity might be attributed to the possibility that GSK2945 activity may be tissue-specific or that it could be regulated by the microenvironment. Furthermore, the various pharmacological activities of GSK2945 can be attributed to circadian phase-dependent modulation.

A 2022 study examined the high-affinity synthetic REV-ERB agonist STL1267 [54]. Interestingly, STL1267 demonstrated a significantly binding mode to REV-ERBs compared to heme. STL1267 showed no adverse or slight effects on cell viability in human hepatocarcinoma HepG2 cells and proliferating C2C12 mouse myoblast cells, respectively. However, although STL1267 alleviated the expression of REV-ERB target gene and circadian modulator, *BMAL1*, in HepG2, no attempts appear to have been made to use STL1267 in the treatment of cancer.

Promising results were obtained in studies with SR8278. Since its first description in 2011 [55], it has been found to demonstrate a number of potentially therapeutic benefits, such as alleviating ferroptosis and renal injuries associated with aristolochic acid nephropathy in mice [56] and promoting corneal wound healing [57]. Despite its structural similarities to the agonist, SR8278 was identified as the REV-ERB inverse agonist. It blocks the ability of the GSK4112 to enhance REV-ERBα-dependent repression in a co-transfection assay. Furthermore, SR8278 upregulates the expression of REV-ERB target genes, such as BMAL1, while GSK4112 downregulates it [55]. SR8278 has been shown to exhibit protective properties in normal tissues exposed to cisplatin; however, the same protective effect may decrease the efficacy of cisplatin against cancer cells [58]. To achieve the desired therapeutic effects in individuals with cancer, it is possible to use this ligand in combination.

### 3.2. Modulators of RORs

The ROR nuclear receptors (full name given in Table 1) regulate the circadian rhythm by acting as transcription factors. The ROR family consists of RORα, RORβ and RORγ. RORs compete with REV-ERBs to modulate the rhythmic expression of the *BMAL1* gene in the core loop, together with other clock-controlled genes [59].

RORs have been found to play regulatory roles in hepatocytes, lung epithelial cells, renal cells and immune cells [60]. A number of studies have shown that RORα expression is significantly decreased during tumor development and progression, and that exogenous RORα expression represses cell proliferation and tumor growth [61]. Table 3 provides a summary and Figure 4 presents the chemical structure of small-molecule ROR ligands investigated in different anticancer therapies.

An example of a natural ligand that directly binds to ROR-alpha and ROR-gamma is nobiletin (NOB), i.e., 5,6,7,8,3′,4′-hexamethoxyflavone: a dietary polymethoxylated flavonoid found in citrus fruits. It has mostly been evaluated as an agent with anti-inflammatory properties that may be of value in preventing metabolic diseases. In addition, other studies indicate that NOB may possess a broad spectrum of mechanisms against cancer development [73]. In one case, treatment was found to inhibit breast cancer cell proliferation by arresting their cell cycle in G1 [74,75]. In another, NOB induced apoptosis of human oral squamous cell carcinoma cell lines (Ca9-22, HSC-3, and TSC-15) in a concentration- and time-dependent manner, while having only a slight effect on primary normal human oral epithelial cells (HOECs) [62].

NOB was also reported to induce G0/G1 cell cycle arrest in ovarian cancer cells [63]. NOB activity may also be related to the circadian clock. In one in vitro study, activation of ROR with NOB treatment appeared to effectively reduce cell growth and inflammation responses in triple-negative breast cancer (TNBC). It was reported that the anticancer and anti-inflammatory effect of NOB is mediated by downregulation of TNF-α and translocation of p65 to the nucleus in the TNBC cells. Moreover, NOB demonstrated a synergistic effect with chemotherapeutic drugs, particularly docetaxel [64].

The first synthetic ligand to bind to and modulate the activity of RORs, i.e., ROR-α and ROR-γ, was T0901317, also known as T1317 [76]. T1317 was found to demonstrate significant activity on liver X receptors (LXRs) and farnesoid X receptors (FXRs), a natural receptor for bile acids; as such, it is not suitable for studying ROR function. However, it was used as a scaffold for derivatives that have been used for work with RORα, RORγ, FXR, LXRα, and LXRβ. One particularly interesting compound was the amide SR1078. Its unique pharmacological profile displays high activity against LXRs and FXRs, making it a good candidate for assessing ROR function [77]. Most of the published work on the use of SR1078 as a ROR modulator is concerned with the regulation of lipid and glucose metabolism [65,78,79], both of which represent promising targets for cancer treatment. One study of the influence of SR1078 on RORα activation in hepatoma cells, whose growth is glutamine dependent [65,80], found treatment to reduce aerobic glycolysis and downregulate biosynthetic pathways in the cells [65]. Similar results were also obtained in gastric cancer cells [66]. Another study examined the effect of SR1078 on RORγ activity in MCF-7 and MDA-MB-231 breast cancer cells: SR1078 was found to significantly decrease cell viability in both cell lines, and 48 to 96 h treatment significantly decreased MCF-7 microsphere outgrowth compared to negative controls (DMSO) [67]. Likewise, SR1078 has also exhibited antitumor activity against ovarian epithelial adenocarcinoma cell lines [68] and neuroblastoma cell lines [69].

Another promising derivative is SR1001, an inverse agonist of ROR-alpha and ROR-gamma, which has mainly been studied as an anti-inflammatory agent. Treatment has been shown to prevent the development of murine T helper 17 cells (TH17) by inhibiting interleukin-17A (IL-17A) gene expression and protein production [81]. As the risk of tumor development is significantly elevated by chronic inflammation [82], other studies have examined the role of the inflammatory Th17-IL-17 pathway in prostate cancer (PC). A study conducted on a mouse model showed that SR1001 treatment significantly decreased the formation of micro-invasive PC [70]. In addition, treatment with the SR1001 derivative SR1555 inhibited TH17 cell development and function; however, it has not been evaluated as an anticancer therapy [83].

Next SR1001 derivative is SR2211, a selective ROR modulator that efficiently inhibits IL-17 production and has shown affinity for ROR-γ [84]. Treatment was found to significantly inhibit proliferation and promote apoptosis in a doxorubicin-resistant prostate cancer cell line, which was attributed to ROR-γ inhibition [71]. SR2211 also demonstrated potential anticancer effects in three androgen receptor-positive tumor models of prostate cancer in mice [72]. Moreover, in vitro studies with MCF-7 cells on the role of ROR-γ in breast carcinogenesis found SR2211 treatment to activate oncogenic pathways (invasion of cells, cell movement and malignancy), but deactivate checkpoint control, DNA repair and homologous recombination [67,72]. The results of pharmacological studies targeting ROR activity with inverse agonists are unclear, which could be due to the possibility that SR2211 activity is tissue-specific. The tumor microenvironment of prostate and breast cancer cells could also play a role in their regulation.

Over the past decades, we have gained a better understanding of the use of circadian modulators in cancer treatment through extensive research. Studies using in vivo and in vitro models indicated a number of important molecular mechanisms for the adjustment of the molecular clock by REV-ERBs and RORs modulators. While many other potential small-molecule modulator compounds are known to exist, including natural (nobiletin) and chemical ligands (SR1078, SR1001, SR2211, SR6452/GSK4112, SR9009, SR9011, STL1267, SR8278), their biological effect on circadian rhythm and impact of tumorigenesis has not yet been fully described [85,86]. The majority of small ligands are agonists, while SR8278, SR1001 and SR2211 are inverse agonists of REV-ERBs and RORs, respectively. Although they may prevent proliferation and metastasis in various cancer types, their precise pharmacological response and mechanisms require deep insight. It is important to evaluate the selectivity of small-molecule ligands that are applied, as the ligand binding domains assign sequence homology to REV-ERB and ROR receptors [85]. However, none of these ligands have been studied as potential modulators of the circadian phase. Only SR9009, SR9011 REV-ERB agonists showed a delay in the initiation of diurnal activity in mice. Thus, synthetic REV-ERB ligands may be useful for the treatment of sleep disorders and also jet lag [84]. Despite these challenges, to evaluate the potential significance of the circadian clock’s pharmacological modulation for cancer treatments, additional research is required. Moreover, the development of research tools and strategies may lead to improved therapies for human cancer treatment.

## Figures and Tables

**Figure 1 ijms-26-11779-f001:**
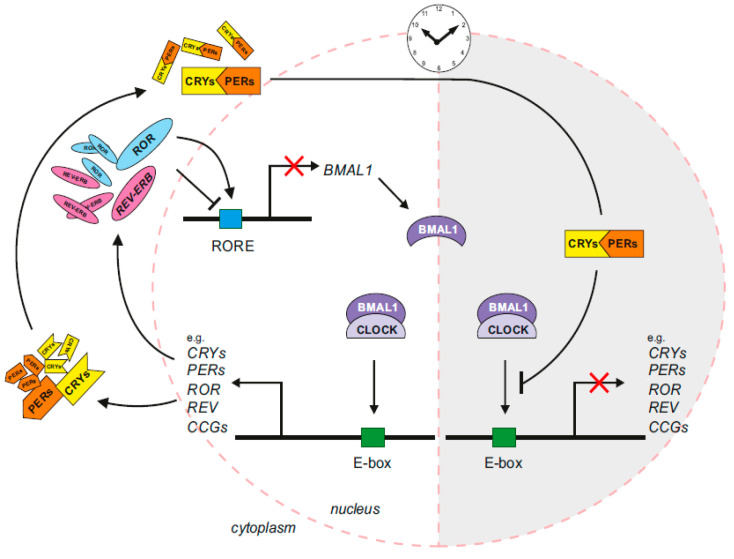
The molecular machinery of the circadian clock scheme (created in CorelDraw Graphics Suite 2021, version: 23.1.0.389).

**Figure 2 ijms-26-11779-f002:**
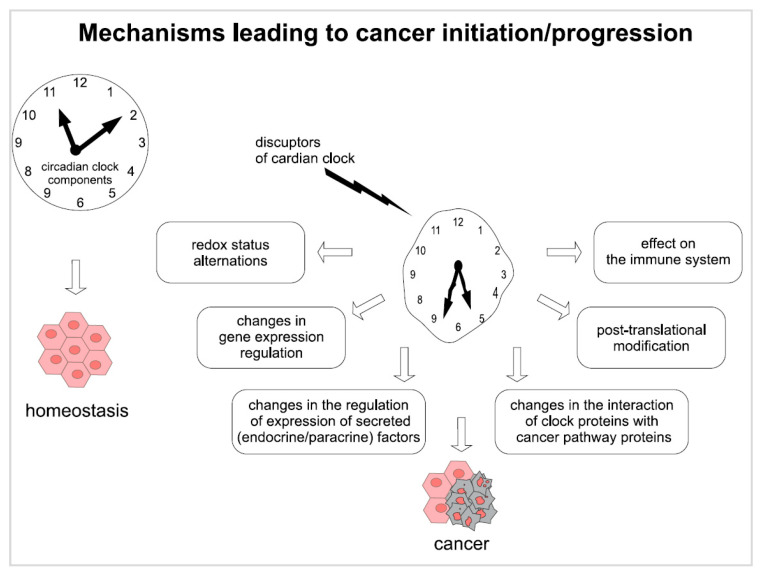
Mechanisms leading to cancer initiation/progression (created in CorelDraw Graphics Suite 2021, version: 23.1.0.389).

**Figure 3 ijms-26-11779-f003:**
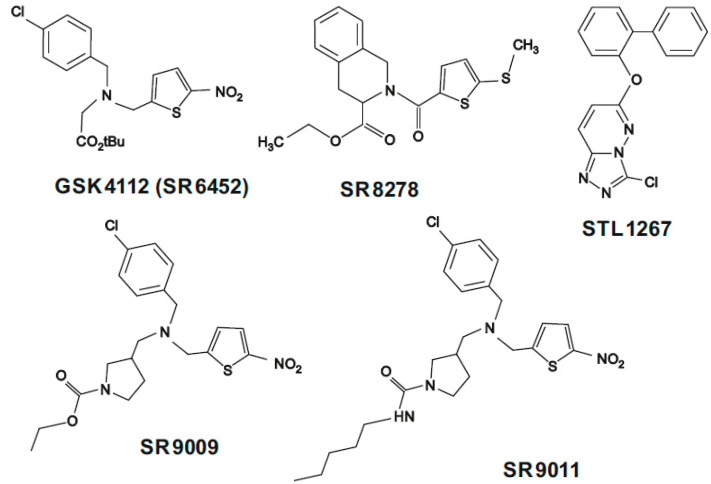
Structure of the small-molecule modulators of REV-ERBs: GSK4112 (SR6452), SR9009, SR9011, STL1267 and SR8278 (created in CorelDraw Graphics Suite 2021, version: 23.1.0.389; database: http://pubchem.ncbi.nlm.nih.gov, accessed on 16 September 2025).

**Figure 4 ijms-26-11779-f004:**
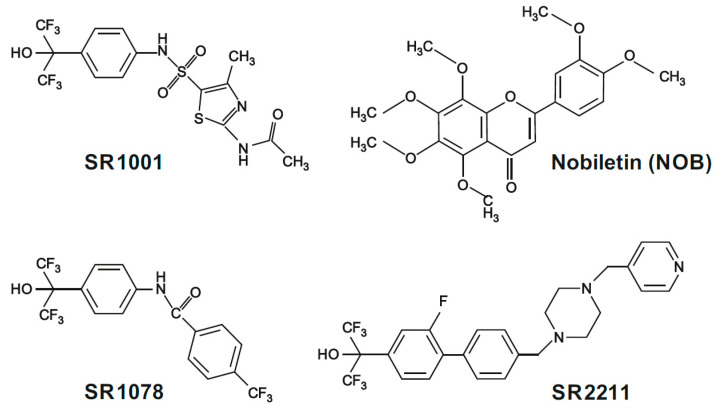
Structure of the small-molecule modulators of RORs: Nobiletin (NOB), SR1078, SR1001 and SR2211 (created in CorelDraw Graphics Suite 2021, version: 23.1.0.389; database: http://pubchem.ncbi.nlm.nih.gov, accessed on 16 September 2025).

**Table 1 ijms-26-11779-t001:** The most important genes forming molecular clocks, i.e., clock genes.

Symbol	Full Name
*NR1D1*	Nuclear receptor subfamily 1 group D member 1
*NR1D2*	Nuclear receptor subfamily 1 group D member 2
*DBP*	D-box binding PAR bZIP transcription factor
*TEF*	TEF, PAR bZIP transcription factor
*HLF*	HLF, PAR bZIP transcription factor
*BHLHE40*	Basic helix–loop–helix family member e40
*BHLHE41*	Basic helix–loop–helix family member e41
*PER1*	Period circadian clock 1
*PER2*	Period circadian clock 2
*PER3*	Period circadian clock 3
*CRY1*	Cryptochrome circadian clock 1
*CRY2*	Cryptochrome circadian clock 2
*RORA*	Retinoic acid receptor (RAR) related orphan receptor A
*RORB*	RAR-related orphan receptor B
*RORC*	RAR-related orphan receptor C
*NFIL3*	Nuclear factor, interleukin 3-regulated
*ARNTL* (*BMAL1*)	Aryl hydrocarbon receptor nuclear translocator-like
*CLOCK*	Clock circadian regulator
*NPAS2*	Neuronal PAS domain protein 2
*CIPC*	CLOCK interacting pacemaker
*CSNK1A1*	Casein kinase 1 alpha 1
*CSNK1A1L*	Casein kinase 1 alpha 1-like
*CSNK1E*	Casein kinase 1 epsilon

**Table 2 ijms-26-11779-t002:** The small-molecule modulators of REV-ERBs.

Compound	Activity	Cancer	Model Research	Effect	References
GSK4112 (SR6452)	Agonist	gastric	SGC-7901 and BGC-823 cell lines	induction of apoptosis	[34]
			BGC-823 and MKN-4 cell lines	induction of ferroptosis	[39]
SR9009	Agonist	small-cell lung	Chemosensitive cells (H69 and H446) and their corresponding chemoresistant cells (H69AR and H446DDP)	significant dose-dependent cytotoxicity induction of apoptosis decreased migration inhibited invasion	[40]
			BALB/c nude mice	antitumor activity independent of the chemoresistance status	
		multiple myeloma	U266 and RPIM8226 cell lines	decline in cell proliferation. induction of caspase 3-mediated apoptosis	[41]
			immune deficient BALB/c nude mice (5 weeks old)	increased antitumor activity of bortezomib, reduced tumor growth, increased survival.	
		bladder	RT-4, T24, and 5637 cell lines	dose-dependent decreased cell viability decreased migration and colony formation	[42]
		hepatocellular	Hepa1c1c7 and HepG2 cell lines	decreased cell viability and migration of Hepa1c1c cells (containing high levels of BMAL1 expression) no effect on HepG2 cells (deficient in BMAL1 expression) and normal (AML12) liver cells	[43]
			HepG2	reduction in cell viability decreased ROS levels increased size of lipid droplet	[44]
		glioblastoma	T98G cells	reduction in cell viability by MTT and alamarBlue assay cells arrested in G0/G1 phases decreased ROS levels increased size of lipid droplet	[44]
		prostate	PC3, 22RV1, DU145, LNCaP, and C4-2B cell lines	no effect on normal prostate cell viability dose-dependent reduction in 22RV1, PC3 and DU145 cell viability, inhibition of colony formation and induction of apoptosis	[45]
			male BALB/c nude mice (6 weeks old)	decreased tumor volume and growth	
SR9011	Agonist	breast	MCF10A, MDA-MB-231, MCF-7, MDA-MB-361, SKBR3, BT474	no effect on non-tumorigenic breast epithelial (MCF-10) cell viability dose-dependent decreased tumorigenic SKBR3 cell viability dose-dependent reduction in cell viability independent of ER, PR or HER2 status (MDA-MB-231, MCF-7, MDA-MB-361, BT474 cell lines) increased in MDA-MB231 cells in the G0/G1 phase	[46]
		osteosarcoma	U2OS, HOS-MNNG, Saos-2 cell lines	suppressed cancer cell growth without effect on normal cells cell cycle arrest in the G0/G1 phase not affect level of the apoptosis regulator BAX	[47]
			female BALB/c nude mice (4–6 weeks old)	decreased tumor volume and growth	

**Table 3 ijms-26-11779-t003:** The small-molecule modulators of RORs.

Compound	Activity	Cancer	Model Research	Effect	References
Nobiletin (NOB)	Agonist	oral squamous cell	Ca9-22, HSC-3 and TSC-15 cell lines	significant dose- and time-dependent cytotoxicity induction of apoptosis of Ca9-22 cell dose-dependent increase in the level of intracellular ROS increased expression of the DNA damage markers γH2 AX and 8-oxodG in Ca9-22 cells in a time-dependent manner	[62]
		ovarian	CaoV3, ES-2, HO-8910, SKOV3, and SKOV3/TAX	reduction in SKOV3/TAX cells growth and proliferation decreased SKOV3/TAX cells colony formation induction of apoptosis through the intrinsic apoptotic pathway impairment of autophagic flux	[63]
		breast	HPNE-P2M, HPNE-Kras, PNAC1, MCF7, MDA-MB-231, MDA-MB-468, DB7 and BT549 cell lines	decreased cell viability in MDA-MB-231, BT549, and MDA-MB-468 cell lines without affecting of MCF10A and MCF7 cell lines reduction in colony formation, colony size and motility in MDA-MB-231 and MDA-MB-468 cells cell cycle arrest in the G2/M phase in MDA-MB-231 cells	[64]
			female FVB mice (5 weeks old)	reduction in tumor size	
SR1078	Agonist	hepatoma	HepG2, Hep3B, and Huh7 cells	inhibition of HepG2 and Hep3B cells proliferation	[65]
		gastric	AGS and MKN-74 cell lines	reduction in colony formation, cell viability, and N-cadherin and Vimentin mRNA levels upregulation of E-cadherin mRNA levels suppression of glycolysis and glycolytic capacity	[66]
		breast	T-47D, MCF-7 and MDA-MB-231 cell lines	reduction in cell viability in both MCF-7 and MDA-MB-231cell lines downregulation of components of the chemokine system (CXCR4 and CXCL12 expression reduction in cell migration in MDA-MB-231 cells	[67]
		ovarian	SKOV-3 and OVCAR-3 cell lines	dose-dependent reduction in cell viability in both cell lines reduced spheroid formation SKOV-3	[68]
		neuroblastoma	SH-SY5Y, IMR32, SK-N-BE(2)-C, SK-N-AS, NGP, LAN5, CHLA255, MYCN3 and NB cell lines	activation of expression of RORα target genes containing a RORE in their promoters (G6Pase, FGF-21, and BMAL1) in both LAN5 and SK-N-BE(2)-C cell lines induction of BMAL1 protein expression in LAN5 cells positive effect on mRNA and BMAL protein levels after activation of MYCN inhibition of cell growth in both MNA and non-MNA cell lines. induction of caspase-mediated apoptosis no effect on p53 protein levels downregulation of genes specifically involved in cholesterol synthesis and FA metabolism in LAN5 cells	[69]
			female athymic Ncr nude mice (4–6 weeks old)	reduction in tumor growth no visible signs of toxicity related to SR1078 treatment sensitization of NB cells to conventional chemotherapy.	
SR1001	Inverse agonist	prostate	*Pten*-null mice (6 weeks old)	reduction in the formation of micro-invasive prostate cancer decreased cellular proliferation increased apoptosis decreases inflammatory cell infiltration decreases angiogenesis upregulation of epithelial marker expression downregulation of mesenchymal markers and transcription factors expression	[70]
SR2211	Inverse agonist	prostate	C4-2B cell line	inhibition of cell growth and proliferation	[71]
			LNCaP, C4-2B, 22Rv1, PC-3, and PC346C	reduction in cell viability and colony formation increased apoptosis downregulation of the key proliferation and survival proteins (e.g., MYC) expression inhibition of androgen receptor signaling	[72]
			male SCID C.B17 mice and BALB/c nu/nu athymic mice (4 weeks old)	inhibition of tumor growth blocked ROR-γ binding to AR-RORE and AR binding inhibition of metastasis to the femur and liver	
		breast	MCF-7	oncogenic effect (activation of invasion of cells, cell movement and malignancy) influence of the TGF-β signaling pathway and DNA-repair module upregulation of the mRNA expression of SMAD3 downregulation of the expression of mRNAs encoding BRCA1 and BRCA2	[67]

## Data Availability

No new data were created or analyzed in this study. Data sharing is not applicable to this article.

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
