# Peer review of "The Circadian Modulators as Molecular Targets in Cancer—A Review"

_ijms, 2025, doi:10.3390/ijms262411779_

Round 1
Reviewer 1 Report
Comments and Suggestions for Authors
The paper of Wolniakowska et al, reviews the link between circadian rhythm disruption and cancer, a complex interplay between the physiology of sleep/wake rhythm regulation, its disruption and “cancer”, although this last term covers too many domains is thus reductionist.
The paper is well written, clear, and documented.
As mentioned by the Authors, the number of papers in this domain is extremely important.
Interestingly (surprisingly?) the term “melatonin” is not cited once (as a solution to cancer). Thank you.
Overall, it is very interesting.
I only have one minor concern, but I guess, the Authors should include a figure with the structures of all the mentioned molecules.
Minor, line 113
It could also be reminded that drug metabolism has a certain circadian rhythm, as it was explored in the 1970/1980 years. This is probably worth mentioning there.
Minor(?) line 123
Of course, as claimed earlier, the downside of this approach is the interplay of the molecule(s) ligand of some of the components of the “rhythm”, and their pharmacological property (agonist, partial agonist, inverse agonist, antagonist and so on !!!). Finding the right modulator could be extremely challenging. Maybe a sentence added there could help the reader, even if in the following sections this problem is mentioned.
Minor, table 2:
As a vaguely interested chemist, I’d suggest that the structure of the molecules cited were put on a figure right after Table 1, and more generally if other compounds are cited – as in the following paragraph.
Minor, Table 3:
Same comment as above
Author Response
The paper of Wolniakowska et al, reviews the link between circadian rhythm disruption and cancer, a complex interplay between the physiology of sleep/wake rhythm regulation, its disruption and “cancer”, although this last term covers too many domains is thus reductionist.
The paper is well written, clear, and documented.
As mentioned by the Authors, the number of papers in this domain is extremely important.
Interestingly (surprisingly?) the term “melatonin” is not cited once (as a solution to cancer). Thank you.
Overall, it is very interesting.
I only have one minor concern, but I guess, the Authors should include a figure with the structures of all the mentioned molecules.
Authors’ response:
Two figures (3 and 4) with the structures of the discussed small molecular modulators have been included in the manuscript as Reviewer has suggested.
Minor, line 113
It could also be reminded that drug metabolism has a certain circadian rhythm, as it was explored in the 1970/1980 years. This is probably worth mentioning there.
Authors’ response:
Thank you for this valuable comment. We have introduced findings regarding possible consequences of the observed daily variations in drug metabolism. Additionally, we have discussed that issue with your next remark (5th page).
Minor(?) line 123
Of course, as claimed earlier, the downside of this approach is the interplay of the molecule(s) ligand of some of the components of the “rhythm”, and their pharmacological property (agonist, partial agonist, inverse agonist, antagonist and so on !!!). Finding the right modulator could be extremely challenging. Maybe a sentence added there could help the reader, even if in the following sections this problem is mentioned.
Authors’ response:
Thank you for this valuable comment. The sentence regarding possible pharmacological activities of the small-molecule modulators has been added (5th page).
Minor, table 2:
As a vaguely interested chemist, I’d suggest that the structure of the molecules cited were put on a figure right after Table 1, and more generally if other compounds are cited – as in the following paragraph.
Authors’ response:
We have introduced adequate structures of the small molecular modulators.
Minor, Table 3:
Same comment as above
Authors’ response:
We have introduced adequate structures of the small molecular modulators.

Reviewer 2 Report
Comments and Suggestions for Authors
Review summarizes recent knowledge of the rapidly evolving area of research at the intersection of circadian biology and oncology. It pays special attention to small-molecule ligands (such as agonists for REV-ERBs and RORs) and their molecular feedback loops in circadian rhythms, that may be used to combat, for example, gastric, breast, prostate, and lung cancers. The authors summarized results of the broad range of studies (e.g., in vitro cell lines, in vivo animal models, and clinical implications). Tables 2 and 3 provide concise summaries of key ligands, including their activities, cancer models, effects (e.g., induction of apoptosis, reduced cell viability), and references. Review balances discussion of natural compounds (e.g., nobiletin from citrus fruits) with synthetic ones (e.g., SR1078, SR2211), providing a holistic view. On the other hand, review seemingly lacks depth on practical challenges like human pharmacokinetics, delivery, ethics, the interplay of circadian modulators with other pathways, and a quantitative comparison to existing therapies.
Overall, this is a useful and informative review that should help researchers interested in circadian pharmacology and oncology and can be recommended.
However, there are several issues that can be addressed for improvement of this review:
-
The review tends to summarize findings without deeper critical analysis or comparisons. For instance, it lists effects (e.g., SR9011's cell cycle arrest in osteosarcoma) but does not discuss limitations, such as variability across cell lines or potential off-target effects. Also as target of these small molecules essentially fluctuate in circadian fashion, one may ask a question, what timing of administration would be optimal? The review does not provide the answers to such question in most cases.
-
For example, some compounds (e.g., GSK2945) are mentioned as showing ambiguous agonist/antagonist behavior (P.7., L.196-197), with microenvironmental factors might influence outcomes. Since review deals with circadian rhythms, possibility of circadian phase (timing)-dependent modulation can be addressed in contexts such as this example.
-
On P.8, L.200, review discusses STL1267 as promising binding factor but lacks detailed anticancer evaluations (e.g., "no attempts appear to have been made to use STL1267 in the treatment of cancer"), leaving a sense of incompleteness.
-
While in vivo studies (e.g., in mice) are cited, human relevance is not deeply explored, and some effects (e.g., SR2211's oncogenic role in breast cancer) introduce contradictions that aren't fully addressed. Furthermore, in some places, the focus looks biased to beneficial effects (e.g., apoptosis induction, reduced tumor growth), with less attention given to drawbacks such as toxicity or lack of specificity. For example, SR8278's protective effects on normal tissues are noted, but its potential to reduce chemotherapy efficacy isn't critically evaluated.
-
The text has minor errors (e.g., "ERV-ERBs" instead of "REV-ERBs", P.5., L.137, inconsistent capitalization, and awkward phrasing like "Studies have found than to alter ..." on P.7., L.160).
Author Response
Review summarizes recent knowledge of the rapidly evolving area of research at the intersection of circadian biology and oncology. It pays special attention to small-molecule ligands (such as agonists for REV-ERBs and RORs) and their molecular feedback loops in circadian rhythms, that may be used to combat, for example, gastric, breast, prostate, and lung cancers. The authors summarized results of the broad range of studies (e.g., in vitro cell lines, in vivo animal models, and clinical implications). Tables 2 and 3 provide concise summaries of key ligands, including their activities, cancer models, effects (e.g., induction of apoptosis, reduced cell viability), and references. Review balances discussion of natural compounds (e.g., nobiletin from citrus fruits) with synthetic ones (e.g., SR1078, SR2211), providing a holistic view. On the other hand, review seemingly lacks depth on practical challenges like human pharmacokinetics, delivery, ethics, the interplay of circadian modulators with other pathways, and a quantitative comparison to existing therapies.
Authors’ response:
We have introduced some of the above-mentioned remarks: daily variation in drug metabolism, the importance of the time of drug administration, etc. (15th page).
Overall, this is a useful and informative review that should help researchers interested in circadian pharmacology and oncology and can be recommended.
However, there are several issues that can be addressed for improvement of this review:
- The review tends to summarize findings without deeper critical analysis or comparisons. For instance, it lists effects (e.g., SR9011's cell cycle arrest in osteosarcoma) but does not discuss limitations, such as variability across cell lines or potential off-target effects. Also as target of these small molecules essentially fluctuate in circadian fashion, one may ask a question, what timing of administration would be optimal? The review does not provide the answers to such question in most cases.
Authors’ response:
Thank you for indicating these very important issues, e.g. study limitations, off-target effects, and timing of administration (9th page).
- For example, some compounds (e.g., GSK2945) are mentioned as showing ambiguous agonist/antagonist behavior (P.7., L.196-197), with microenvironmental factors might influence outcomes. Since review deals with circadian rhythms, possibility of circadian phase (timing)-dependent modulation can be addressed in contexts such as this example.
Authors’ response:
The suggested modulation of an effect by a putative phase-dependent modulation has been discussed (9th page).
- On P.8, L.200, review discusses STL1267 as promising binding factor but lacks detailed anticancer evaluations (e.g., "no attempts appear to have been made to use STL1267 in the treatment of cancer"), leaving a sense of incompleteness.
Authors’ response:
We have thoroughly discussed the pointed issue (9th page).
- While in vivo studies (e.g., in mice) are cited, human relevance is not deeply explored, and some effects (e.g., SR2211's oncogenic role in breast cancer) introduce contradictions that aren't fully addressed. Furthermore, in some places, the focus looks biased to beneficial effects (e.g., apoptosis induction, reduced tumor growth), with less attention given to drawbacks such as toxicity or lack of specificity. For example, SR8278's protective effects on normal tissues are noted, but its potential to reduce chemotherapy efficacy isn't critically evaluated.
Authors’ response:
We have thoroughly discussed the pointed issues (10th and 14th/15th pages)
- The text has minor errors (e.g., "ERV-ERBs" instead of "REV-ERBs", P.5., L.137, inconsistent capitalization, and awkward phrasing like "Studies have found than to alter ..." on P.7., L.160).
Authors’ response:
We appreciate you pointing out these errors and we have made the necessary corrections.
